# Treatment of Calcific Insertional Achilles Tendinopathy: Knotless Internal Brace versus Knot-Tying Suture Bridge

**DOI:** 10.3390/jpm13030404

**Published:** 2023-02-24

**Authors:** Xiaodong Zhao, Xiaolei Yang, Yifan Hao, Fujun Yang, Zhenping Zhang, Qirong Qian, Peiliang Fu, Qi Zhou

**Affiliations:** 1Department of Sports Medicine, Weifang Hospital of Traditional Chinese Medicine, Weifang 261021, China; 2Department of Anesthesia, Naval Medical University Second Affiliated Hospital, Shanghai 200003, China; 3Department of Orthopedics, Naval Medical University Second Affiliated Hospital, Shanghai 200003, China

**Keywords:** suture bridge, internal brace, calcific Achilles tendinopathy

## Abstract

Background: This study aimed to compare the knotless internal brace technique and the knot-tying suture bridge technique via the medial approach in the treatment of calcific Achilles tendinopathy. Methods: The clinical data of 25 cases of calcific Achilles tendinopathy in which nonoperative treatments had failed were retrospectively collected. All the patients received Achilles tendon debridement and Haglund deformity excision through a medial approach, followed by repair using the knotless internal brace technique or the knot-tying suture bridge technique. Pain was evaluated by using the visual analog scale (VAS). The American Orthopedic Foot and Ankle Score (AOFAS) questionnaire was administered preoperatively and postoperatively. Results: The mean follow-up time was 2.6 (range 2–3.5) years. There were no wound complications and no Achilles tendon ruptures. At 1 year postoperatively, the internal brace group was superior to the suture bridge group in terms of the VAS scores (*p* = 0.003). However, no differences were noticed between the two groups in either the VAS or the AOFAS scores at 2 years postoperatively. Conclusions: The medial approach in combination with the suture bridge technique was effective in treating calcific Achilles tendinopathy. The knotless internal brace technique involved less pain compared to the knot-tying suture bridge technique only at the early postoperative stage.

## 1. Introduction

Posterior heel pain is a common clinical problem and is seen in up to 15% of patients presenting to their primary care clinicians [1]. It is usually caused by soft-tissue or osseous abnormalities, such as insertional Achilles tendinopathy, Haglund deformity, retrocalcaneal bursitis and intratendinous or insertional calcifications of the Achilles tendon. Soft-tissue and osseous abnormalities often coexist. Intratendinous calcific lesions of the Achilles tendon often cause more severe pain and exert a greater impact on daily life.

Several theories have been proposed for the development of calcification in the Achilles insertion or midportion [2,3]. In one of these theories, the repetitive movement of the Achilles tendon leads to the stripping of the periosteum around the Achilles insertion, which results in subperiosteal bleeding and calcification. Another theory suggests that movement of the Achilles tendon results in subcartilaginous osseous metaplasia and calcification. Benjamin et al. suggested that the insertional calcification of the Achilles tendon is not induced by inflammation and microtears but may increase the interface between the tendon and the bone [4].

Poor results after at least 6 months of conservative management for heel pain indicate the requirement of surgical treatments, such as Haglund deformity excision, debridement of the calcific lesions, and calcaneal exostectomy [5]. The thorough debridement and repair of the Achilles insertion are critical for good treatment outcomes after either partial or complete Achilles tendon detachment. The Achilles insertion is usually repaired by reattaching the tendon to the calcaneus using anchors and sutures. Suture anchors include single-row and double-row repair, which each have advantages and disadvantages in biomechanical properties [6,7]. The double-row suture bridge technique has been increasingly used in recent years. This technique uses crossing suture bridges to obtain a greater area of tendon compression and greater stability of the Achilles insertion [8].

Double-row suturing is usually performed with the suture bridge technique, including the internal brace technique and the knot-tying/knotless suture bridge technique [9]. These techniques are widely used in repairing the rotator cuff and have similar mid-term or long-term results [10]. However, internal bracing is thought to be advantageous for rotator cuffs with weak tendons due to the lower risk of retears [11]. These techniques have also been used in repairing the Achilles tendon. A biomechanical study suggested that knot-tying suture bridge repair has a higher load to failure compared with the knotless suture bridge technique [12]. However, this has not been corroborated by clinical studies.

The present study aimed to retrospectively compare the treatment outcomes of calcific Achilles tendinopathy between the knotless internal brace technique and the knot-tying suture bridge technique.

## 2. Materials and Methods

### 2.1. Patients

This retrospective study screened all the patients who presented with Achilles disorders to our hospital from January 2013 through January 2020. The inclusion criteria included a diagnosis of insertional Achilles tendinopathy or Haglund syndrome, age > 18 years, failed nonoperative treatment ≥ 6 months, radiographic findings of intratendinous or insertional calcifications of the Achilles tendon, and Achilles tendon repair using the knot-tying suture bridge technique (before 2017) or the knotless internal brace technique (after 2017). Patients with the following conditions were excluded: previous hindfoot trauma or surgery; rheumatoid or infectious arthritis; gout, tuberculosis, malignant tumor, or Charcot’s joint; neuromuscular disorders of the affected limb; diabetes mellitus; incomplete follow-up data.

This study was approved by the ethics committee of our hospital. Informed consent was waived due to the retrospective nature of the study.

### 2.2. Surgical Procedure

All operations were performed by a single surgeon (Q.Z.) using general anesthesia and with the patient in the prone position. Cefazolin sodium (2 mg/kg) was intravenously administered half an hour preoperatively, and clindamycin was used instead upon allergy to cephalosporins. A pneumatic tourniquet was applied at the thigh level. A medial incision in the Achilles tendon was made. A second transverse incision was made at the insertion of the Achilles tendon. Next, the two incisions were connected by a curved incision. The skin and subcutaneous tissue were dissected, and the Achilles tendon aponeurosis was exposed. The aponeurosis was incised longitudinally along the medial edge of the Achilles tendon and sutured onto the subcutaneous tissue. The dissection was performed laterally underneath the aponeurosis to fully expose the Achilles tendon. At the insertion level, the Achilles tendon was split centrally. The insertion was detached, keeping about 30% to 40% of the insertional area. With the ankle joint in plantar flexion, the posterior bursa of the Achilles tendon was exposed and resected. An osteotomy of the posterosuperior osteophyte of the calcaneus was performed. The bony edge was filed to make it smooth. Complete osteotomy was guaranteed by fluoroscopy. The calcific component of the Achilles tendon was located by palpating its anterior surface and debrided from the healthy tissue.

After thorough irrigation with normal saline, the Achilles tendon was repaired using either the knotless internal brace technique or the knot-tying suture bridge technique. For internal bracing, the medial row screws used the Arthrex AR-2324BCCT BioComposite SwiveLock C (4.75 mm × 19.1 mm) and the lateral technique used the Arthrex AR-2324BCC BioComposite SwiveLock C (4.75 mm × 19.1 mm) (Figure 1). For suture bridge, the medial row screws used the Smith & Nephew TWINFIX Ultra PK (4.5 mm) and the lateral used the Arthrex AR-2324BCC BioComposite SwiveLock C (4.75 mm × 19.1 mm) (Figure 2). The Achilles tendon and the soft tissues were closed using 2-0 absorbable sutures.

### 2.3. Postoperative Management

The ankle joint was positioned in plantar flexion at 20° and fixed with a plaster cast for two weeks. After the removal of the cast, an adjustable brace was used. Plantar flexion of no more than 20° was allowed with no weight-bearing of the operated leg. Considering the high tension of the calcaneus skin, the sutures were removed three weeks postoperatively. Next, the patients were encouraged to bear their weight with double elbow crutches. Three layers of heel soles were used with the ankle brace, with one layer removed per week. Full range of motion of the ankle was allowed six weeks postoperatively. The ankle brace was worn until eight weeks postoperatively.

### 2.4. Patient Assessment

All patients were assessed for pain and ankle function preoperatively, 1 year, and 2 years postoperatively. Pain was evaluated by using the visual analog scale (VAS). In addition, the patients completed the Victorian Institute of Sport Assessment-Achilles questionnaire and the American Orthopedic Foot and Ankle Score (AOFAS) questionnaire.

### 2.5. Statistical Analysis

Numerical variables were expressed as mean and standard deviation. Data normality was tested using the Shapiro–Wilk test. Count variables were expressed as numbers and percentages. Comparisons were made between patients treated with the internal brace technique and the suture bridge technique. All statistical analyses were performed using SPSS (19.0, IBM, Armonk, NY, USA). A *p*-value of less than 0.05 was considered statistically significant.

## 3. Results

A total of 25 patients (25 heels) met the inclusion and exclusion criteria and were analyzed. The mean follow-up time was 2.6 (range 2–3.5) years. The internal brace technique was used in 12 patients and the suture bridge technique in 13 patients. The mean age was 54.1 (range 37–75) years. There was no significant difference in the general characteristics between the two groups (Table 1).

The internal brace group and the suture bridge group showed no significant differences in their VAS scores and AOFAS scores at baseline. At 1 year postoperatively, the internal brace group was superior to the suture bridge group in terms of VAS scores (Table 2). However, no differences were noticed in the VAS scores between the two groups at 2 years postoperatively. The AOFAS scores showed no significant difference between the two groups at 1 and 2 years postoperatively. In addition, each group had significant improvements in both pain and ankle function at 1 year and 2 years postoperatively compared with the baseline.

Lateral radiographs of the ankle 1 day postoperatively showed the complete removal of the calcific lesions in the Achilles tendon (Figure 3 and Figure 4). No loosening or fracture of the anchors occurred during the follow-up. All the surgical wounds healed well with no infection, scar, or dehiscence. No severe complications occurred, such as sural nerve injury, saphenous vein injury, or venous thrombosis. No patients experienced Achilles tendon rupture, contracture, or adhesion. There was no significant difference in the range of motion between the affected ankles and the contralateral ones (dorsiflexion: 23.7 ± 1.5 vs. 23.8 ± 1.9 degrees, *p* = 0.503; plantar flexion: 38.8 ± 2.2 vs. 39.3 ± 2.1, *p* = 0.080). At the last appointment, all 25 patients were asymptomatic and had returned to their pre-injury levels of activity and mild sports.

## 4. Discussion

There is still some controversy over the proper extent of the debridement of the Achilles insertion in the surgical treatment of calcific Achilles tendinopathy. It has been proposed that residual tendinous tissue after the partial detachment of the Achilles insertion may lead to biomechanical instability, incomplete pain relief, relapsed calcific lesions, and even Achilles tendon rupture [13,14,15]. Compared with longitudinal splitting and detachment, the complete detachment of the Achilles insertion is timesaving and more thorough in the debridement of calcific lesions. However, in 95% of patients with calcific Achilles tendinopathy, the calcific lesions are in the middle third of the Achilles tendon [16]. Therefore, a 70% detachment of the Achilles tendon is adequate for the thorough debridement of calcific lesions and diseased tendinous tissue, eliminating the need for the complete detachment of the Achilles insertion [17]. In our patients, the Achilles insertion was detached by more than 50%, but not completely. The surgical procedures were not hindered by the medial or lateral residual tendinous insertional sites, and the calcific lesion debridement was complete. No persistent pain or Achilles tendon retear occurred in our patients during the follow-up.

Kolodziej et al. suggested that the detachment of over 50% of the Achilles insertion should be repaired, otherwise there is a risk of complete avulsion [18]. The repair of the Achilles insertion can be performed by using the single-row or knot-tying double-row suture bridge technique, or the knotless double-row suture bridge technique [19]. However, there is still no conclusive evidence concerning which technique has the best biomechanical properties and treatment outcomes. It has been shown that single-row repair is similar to the double-row suture bridge technique in terms of load to failure and cyclic displacement, which is not significantly associated with the size of the rotator-cuff rupture [20]. A biomechanical study found that the Achilles insertion repaired by the double-row technique had less displacement during cyclic loading but could not stand more load before clinical failure than single-row suture anchoring [6]. However, a meta-analysis consisting of eight biomechanical studies suggested that double-row repair is superior to single-row repair in load to failure and tear resistance in rotator cuffs [21]. The use of the suture bridge technique has been expanded from rotator-cuff repair to the repair of other tendons and ligaments, including the Achilles insertion [13,22]. Byrne et al. reported a case of an elite athlete who returned to competition 18 weeks after the repair of their Achilles tendon rupture using knotless internal bracing [23]. A cadaveric study suggested that the knot-tying suture bridge has a significantly higher load to failure compared with the knotless suture bridge in repairing the Achilles insertion [12]. A retrospective study with 38 patients with insertional Achilles tendinopathy showed that knotless and knot-tying double-row repair techniques have similar treatment outcomes, complications, and pain [24]. In addition, knotless internal bracing is considered advantageous, with less tissue ischemia and infection, and fewer granulomas [25,26,27].

Another concern about knot-tying in the suture bridge technique is tendon healing and retear. Knot-tying can presumably reduce the blood flow in the tendon. A study found that knot-tying is associated with the incomplete healing of the rotator cuff 24 months post-operatively. However, there were no significant differences in clinical outcomes and retear rates between the techniques with or without knot-tying at 3, 12, and 24 months postoperatively [28]. Similarly, another study found no significant difference in the retear rate of the rotator cuff between the knotless and knot-tying suture bridge techniques at 6 months postoperatively when using ultrasound or magnetic resonance imaging [25]. We also found no retear of the Achilles tendon in our patients during the follow-up, which is consistent with the results of Scott et al. [24]. The risk of retear after Achilles repair when using the suture bridge technique should be further investigated.

Our study did not include patients with diabetes, although it was previously found that diabetic patients had more postoperative pain after Achilles tendon repair [24]. In our study, the knotless internal brace technique was superior to knot-tying suture bridge anchoring in terms of VAS scores at postoperative 1 year, but not at two years. Furthermore, the two groups of patients showed no significant difference in terms of their AOFAS scores, suggesting that knot-tying in the suture bridge technique might have no significant effect on postoperative tendon function. After all, tendon recovery is primarily determined by tissue regeneration, and the suturing technique is only a secondary factor. Knot-tying may increase the tension in the Achilles tendon and result in more postoperative pain. This pain is usually relieved with time after the tension reduces. Furthermore, the suture knots on the Achilles tendon may cause irritation and pain in the subcutaneous tissue. The pain may decrease after the suture knots are wrapped up by the scar tissue.

A posterior median approach has good visualization but may also result in many complications. Gillis et al. [27] regarded posterior scar as one of the major disadvantages of the central tendon-splitting approach. Therefore, skin wound healing and the prevention of infections are clinical challenges no matter which approach is used. The medial J-approach is free of the posterior scare associated with the central tendon-splitting approach [14]. However, the J-approach is prone to poor wound healing, especially at the junction of the incisions. A high risk of poor wound healing is noted if the junction angle is less than 90°, in our experience. To reduce this risk, we used a curved incision to connect the longitudinal incision and the transverse incision. None of our patients had wound-healing issues or long-term scars causing discomfort.

Our study features some limitations. Firstly, this was a retrospective study and lacked randomization in the patient assignment. Secondly, the sample size was small due to the scarcity of patients with calcific Achilles tendinopathy. Thirdly, the strict exclusion criteria might reduce the representativeness of the patient sample. Fourthly, the follow-up time was relatively short. Our future investigation should focus on the long-term treatment results with a larger sample size and a randomized study design.

In conclusion, surgery is indicated after failed nonoperative treatments for calcific Achilles tendinopathy. The medial approach in combination with knotless internal bracing or knot-tying suture bridge anchoring is an effective surgical method for Achilles tendon repair. Internal bracing involved less pain than knot-tying suture bridge anchoring at the early postoperative stage. The two techniques were similar in terms of postoperative ankle function and complications.

## Figures and Tables

**Figure 1 jpm-13-00404-f001:**
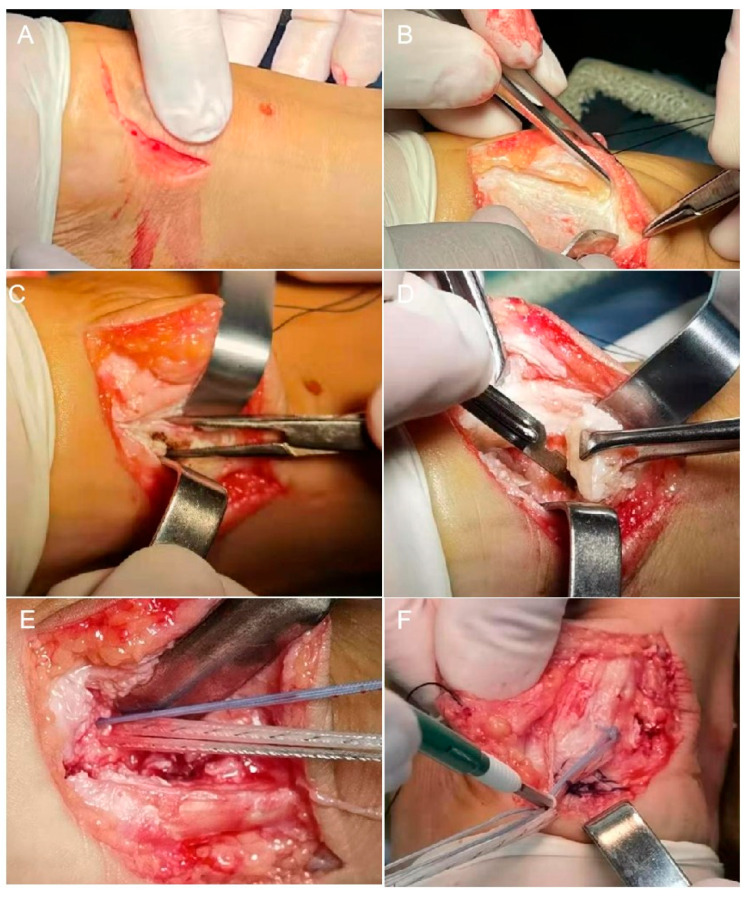
Intraoperative photographs of internal bracing. (**A**) Surgical incision. (**B**) The Achilles tendon was exposed. (**C**) Splitting the Achilles tendon through the midline. (**D**) Resection of the calcific lesion and the posterior bursa of the Achilles tendon. (**E**) Implantation of the medial row screw. (**F**) Implantation of the lateral row screw.

**Figure 2 jpm-13-00404-f002:**
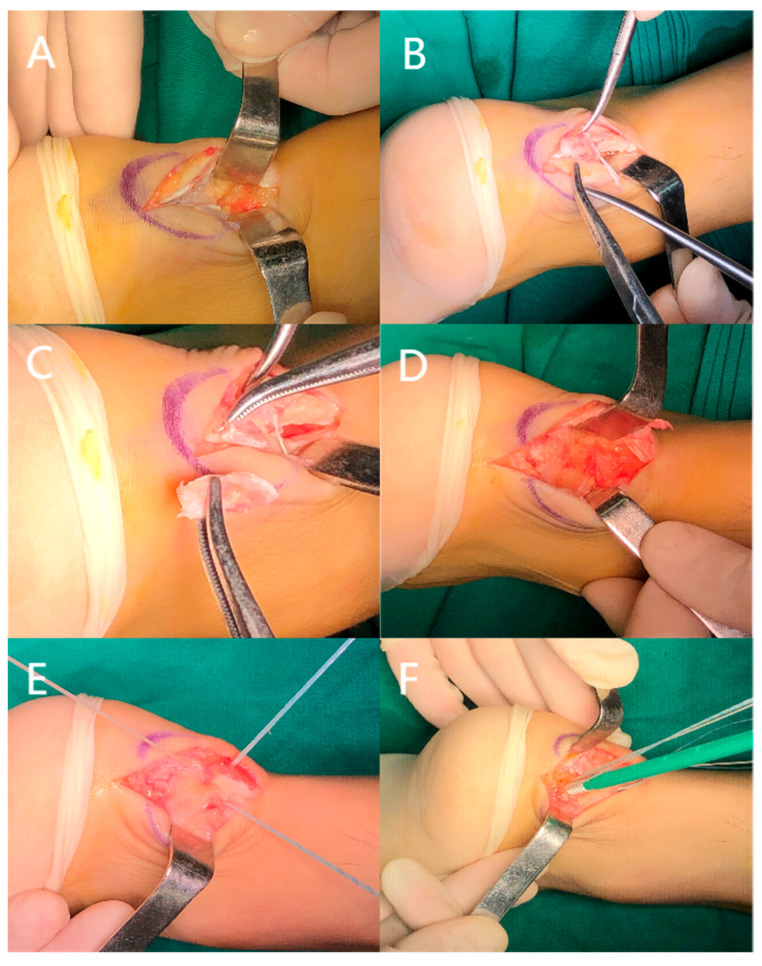
Intraoperative photographs of knot-tying suture bridge anchoring. (**A**) Surgical incision and exposure of the Achilles tendon. (**B**) The lesion was exposed. (**C**) Resection of the calcific lesion. (**D**) Resection of the bursa and osteophyte. (**E**) Implantation of the medial row screw and tying of the knots. (**F**) Implantation of the lateral-row screw.

**Figure 3 jpm-13-00404-f003:**
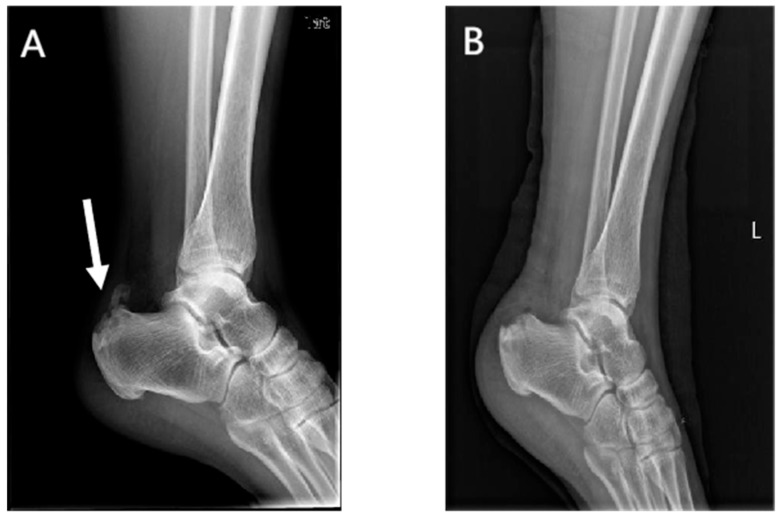
Preoperative (**A**) and postoperative (**B**) lateral radiographs of the ankle in the internal brace group. Arrow, a calcific lesion in the Achilles tendon.

**Figure 4 jpm-13-00404-f004:**
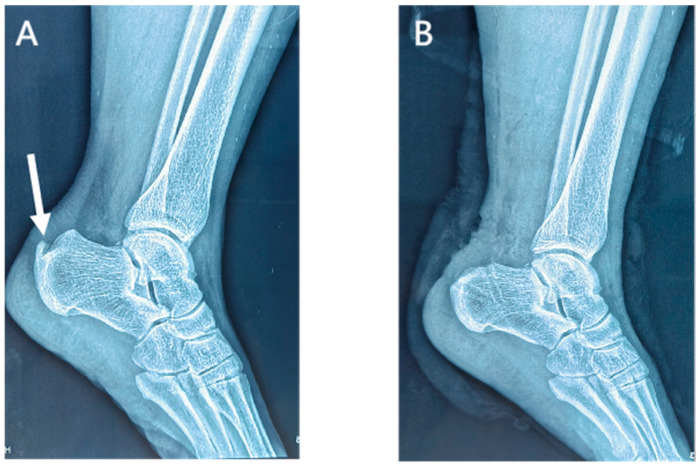
Preoperative (**A**) and postoperative (**B**) lateral radiographs of the ankle in the suture bridge group. Arrow, a calcific lesion in the Achilles tendon.

**Table 1 jpm-13-00404-t001:** General characteristics of the patients.

	Internal Brace (n = 12)	Suture Bridge (n = 13)	*p*-Value
Age, yr	54.9 ± 9.5	53.4 ± 9.5	0.69
Male, n (%)	8 (66.7)	8 (61.5)	0.56
Body mass index, kg/m^2^	29.3 ± 4.0	27.2 ± 2.9	0.15
Left limb affected, n (%)	4 (33.3%)	5 (38.5)	0.56
Fowler-Philip angle, degree	73.2 ± 5.3	69.8 ± 6.3	0.16
Follow-up time, yr	2.6 ± 0.64	2.6 ± 0.65	0.97
Occupation, n			0.86
Manual worker	6	6	
White-collar	3	4	
Driver	1	2	
Other	2	1	

**Table 2 jpm-13-00404-t002:** Assessment of pain and ankle function.

	Internal Brace (n = 12)	Suture Bridge (n = 13)	*p*-Value
VAS scores			
Preoperative	8.25 ± 0.75	7.77 ± 0.83	0.145
Postoperative 1 year	1.92 ± 0.79 ^#^	3.08 ± 0.95 ^#^	0.003
Postoperative 2 years	1.67 ± 0.78 ^#^	1.54 ± 0.66 ^#^	0.66
AOFAS scores			
Preoperative	50.3 ± 5.89	50.1 ± 3.88	0.931
Postoperative 1 year	89.3 ± 4.96 ^#^	86.9 ± 3.3 ^#^	0.176
Postoperative 2 years	90.4 ± 4.8 ^#^	89.4 ± 3.2 ^#^	0.53

VAS: visual analog scale; AOFAS: American Orthopedic Foot and Ankle Score. # vs. preoperative, *p* < 0.05.

## Data Availability

The data presented in this study are available on request from the corresponding author. The data are not publicly available due to institutional regulations.

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
