# Peer review of "Treatment of Calcific Insertional Achilles Tendinopathy: Knotless Internal Brace versus Knot-Tying Suture Bridge"

_jpm, 2023, doi:10.3390/jpm13030404_

Round 1

Reviewer 1 Report

The manuscript is well written and the methods reasonable, given the relative rarity of the condition treated. 

There is a typo on line 191 that should be fixed (around the word "easy") and the next sentence is somewhat unclear.

Author Response

The manuscript is well written and the methods reasonable, given the relative rarity of the condition treated.

There is a typo on line 191 that should be fixed (around the word "easy") and the next sentence is somewhat unclear.

►Answer: Thanks for the comment. This part of the discussion is deleted because it is not closely related to the research topic. The revised discussion now focuses on comparison of the two repair methods, knotless internal bracing vs. knot-tying suture bridge, for Achilles tendinopathy.

Reviewer 2 Report

I commend you on your submission, and thank you for your efforts.   I do think it is very important to look at these types of constructs not just in the cadaveric model, but in patients to determine which really is the optimal configuration.Unfortunately, I have some concerns with the methodology of this paper that I think may limit its act applicability.

Line by line issues and manuscript critique:

TITLE: Does the title clearly portray the subject and purpose of the study?

 Title is fine.

    ABSTRACT: Does the abstract accurately reflect the study? Are all pertinent finds included?

How were the cohorts assigned? Was there a randomization protocol? Obviously this is subject to a lot of bias if you were determining which repair type that you were going to use based specifically on intra operative findings.

    INTRODUCTION: What is the authors' original research question, and does their study support or fulfill it?

Line 28 – odd use of “outpatients”

Line 40 – do not use a contraction like “doesn’t” in scientific writing.

In general, the introduction requires revision, as currently it reads poorly

Line 43 -  I would certainly not say that based on a cadaveric study and a study looking at early weight bearing that you could say the results OVERALL are better of the double row. This is a gross overstatement.

Line 48 – cite

    METHODS: Was the research method or study design appropriate? Is it presented sufficiently so that other researchers can duplicate them? Are the sample sizes adequate? Are the statistical analyses appropriate and correct?

Line 106 – “planta flexion”

These methodologies are incomplete. There is no mention whatsoever of any sort of randomization protocol. How was the decision made to proceed with one type of repair versus another?

-was a power analysis done??

    RESULTS: Do the results answer the original research questions, as demonstrated in the Results section and tables and figures?

The results are not presented completely. It is important of course to search for differences between the cohorts, but the authors have not included important factors such as BMI and medical comorbidities.

As I would expect, there were no differences ultimately between the two repair types at final follow up. I do think it is actually a somewhat interesting finding that there was a little bit more pain at the early follow up in the knot group, which makes sense that maybe patients are bothered by the subcutaneous knots. But otherwise all this study serves to do is further corroborate that there are probably few differences in rerupture and complications between knot less constructs and knotted constructs.

    DISCUSSION: Is the Discussion balanced? Does it put the results in context? Do the authors acknowledge the limitations of the study?

182- 211 - why are you focusing so much on the approach used for Achilles tendon debridement and describing this much of the anatomy? The study is comparing two different suture constructs for Achilles tendon repair after Haglund’s debridement, the discussion needs to be about that. All this other information is completely superfluous here.

Overall, I think the discussion does not spend enough time discussing the actual topic of the paper. You need to include other relevant literature discussion such as below:

https://pubmed.ncbi.nlm.nih.gov/25225681/

https://pubmed.ncbi.nlm.nih.gov/33030070/

The whole discussion should basically be rewritten.

    CONCLUSIONS: Are the conclusions supported by the study findings? Does the study provide new, unique, or confirmatory findings? Will the findings be of interest to clinicians or to the public?

Conclusions are generally fair, although it should be stated that results in general were not better at the early term, only the VAS pain score was.

    TABLES AND FIGURES: Are all data presented in the text and tables and figures consistent? Do the tables clearly present information not easily summarized in the text of the paper? Are all of the tables necessary? Are the figures necessary and appropriate? Are they of high quality and clearly labeled? Can any be deleted?

Tables are incomplete, need more patient demographic variables.

    REFERENCES: Is the References section complete, or is it excessive? Does it include all of the necessary current, relevant sources

No -- more literature needs to be cited, more cadaveric studies need to be discussed. You need to spend much of the discussion actually discussing the topic at hand here.

Author Response

I commend you on your submission, and thank you for your efforts. I do think it is very important to look at these types of constructs not just in the cadaveric model, but in patients to determine which really is the optimal configuration. Unfortunately, I have some concerns with the methodology of this paper that I think may limit its act applicability.

Line by line issues and manuscript critique:

TITLE: Does the title clearly portray the subject and purpose of the study?

Title is fine.

ABSTRACT: Does the abstract accurately reflect the study? Are all pertinent finds included?

How were the cohorts assigned? Was there a randomization protocol? Obviously this is subject to a lot of bias if you were determining which repair type that you were going to use based specifically on intra operative findings.

►Answer: This is a retrospective study analyzing preexisting clinical routine data between January 2013 and January 2020. No randomization was used to assign the patients to the treatments. The knotless internal brace technique was introduced at our hospital in 2017. Therefore, all patients treated before that time were managed with the knot-tying suture bridge technique, and those after were mostly managed with the knotless internal brace technique. We have discussed this as a limitation.

INTRODUCTION: What is the authors' original research question, and does their study support or fulfill it?

Line 28 – odd use of “outpatients”

►Answer: Thanks for the suggestion. This has been changed to “patients presenting to their primary care clinicians”.

Line 40 – do not use a contraction like “doesn’t” in scientific writing.

►Answer: Thanks for the suggestion. This part has been removed.

In general, the introduction requires revision, as currently it reads poorly.

►Answer: Thanks for the suggestion. We have rewritten the Introduction.

Line 43 -  I would certainly not say that based on a cadaveric study and a study looking at early weight bearing that you could say the results OVERALL are better of the double row. This is a gross overstatement.

►Answer: Thanks for the suggestion. We have revised the Introduction to improve the accuracy of the description of single-row versus double-row repair (the third paragraph of the Introduction).

Line 48 – cite

►Answer: A new reference has been added. “However, internal bracing is thought to be advantageous for rotator cuff with weak tendons due to lower risk of retears [10].”

METHODS: Was the research method or study design appropriate? Is it presented sufficiently so that other researchers can duplicate them? Are the sample sizes adequate? Are the statistical analyses appropriate and correct?

Line 106 – “planta flexion”

►Answer: It has been changed to “perform planta flexion of no more than 20°”.

These methodologies are incomplete. There is no mention whatsoever of any sort of randomization protocol. How was the decision made to proceed with one type of repair versus another?

►Answer: This is a retrospective study analyzing preexisting clinical routine data between January 2013 and January 2020. No randomization was used to assign the patients to the treatments. The knotless internal brace technique was introduced at our hospital in 2017. Therefore, all patients treated before that time were managed with the knot-tying suture bridge technique, and those after were mostly managed with the knotless internal brace technique. We have discussed this as a limitation.

-was a power analysis done??

►Answer: The sample size was not calculated but was decided according to previous similar studies.

RESULTS: Do the results answer the original research questions, as demonstrated in the Results section and tables and figures?

The results are not presented completely. It is important of course to search for differences between the cohorts, but the authors have not included important factors such as BMI and medical comorbidities.

►Answer: Thanks for the suggestion. We have added the BMI and occupation data in Table 1. Although previous research suggested that BMI may have impact on treatment outcomes of heel pain, we found no significant difference in BMI between the two groups. To minimize the effect of comorbidities on the surgical results, we excluded patients with previous hindfoot trauma or surgery, rheumatoid or infectious arthritis, gout, tuberculosis, malignant tumor, Charcot’s joint, neuromuscular disorders of the affected limb, or diabetes mellitus. Therefore, we present no data of comorbidities.

As I would expect, there were no differences ultimately between the two repair types at final follow up. I do think it is actually a somewhat interesting finding that there was a little bit more pain at the early follow up in the knot group, which makes sense that maybe patients are bothered by the subcutaneous knots. But otherwise all this study serves to do is further corroborate that there are probably few differences in rerupture and complications between knot less constructs and knotted constructs.

►Answer: The knot-tying suture bridge technique was associated with more pain in the early postoperative phase. This could be related to the subcutaneous knots or the uneven tension of the knots. We have revised the manuscript with inspirations from your comment.

DISCUSSION: Is the Discussion balanced? Does it put the results in context? Do the authors acknowledge the limitations of the study?

182- 211 - why are you focusing so much on the approach used for Achilles tendon debridement and describing this much of the anatomy? The study is comparing two different suture constructs for Achilles tendon repair after Haglund’s debridement, the discussion needs to be about that. All this other information is completely superfluous here.

Overall, I think the discussion does not spend enough time discussing the actual topic of the paper. You need to include other relevant literature discussion such as below:

https://pubmed.ncbi.nlm.nih.gov/25225681/

https://pubmed.ncbi.nlm.nih.gov/33030070/

The whole discussion should basically be rewritten.

►Answer: Thanks for the suggestion. We have written the discussion and removed the less relevant contents. The focus now is on the comparison between the two repair methods. We kept the discussion on skin incision and protection of the soft tissues. Afterall, skin necrosis and wound non-healing are challenging issues no matter which repair method is used.

CONCLUSIONS: Are the conclusions supported by the study findings? Does the study provide new, unique, or confirmatory findings? Will the findings be of interest to clinicians or to the public?

Conclusions are generally fair, although it should be stated that results in general were not better at the early term, only the VAS pain score was.

TABLES AND FIGURES: Are all data presented in the text and tables and figures consistent? Do the tables clearly present information not easily summarized in the text of the paper? Are all of the tables necessary? Are the figures necessary and appropriate? Are they of high quality and clearly labeled? Can any be deleted?

Tables are incomplete, need more patient demographic variables.

►Answer: Thanks for the suggestion. We have added the BMI and occupation data in Table 1.

REFERENCES: Is the References section complete, or is it excessive? Does it include all of the necessary current, relevant sources

No -- more literature needs to be cited, more cadaveric studies need to be discussed. You need to spend much of the discussion actually discussing the topic at hand here.

►Answer: Thanks for the suggestion. We have written the discussion.

Round 2

Reviewer 2 Report

Thank you for your time in responding to my comments, I think the manuscript has been much improved. However, I still have a number of concerns with this manuscript. Some of my comments you have not addressed at all, for example, in the conclusion you state “internal bracing only showed superiority to knot-tying suture bridge anchoring at the early postoperative stage” and you do not mention that this is only with regards to VAS pain score. I really think that is a result that needs to be greatly qualified, and I am concerned that you did not address this point when I had brought it up on my previous review.

Furthermore, when you attempted to address my concern about planta flexion, you completely missed my point, the correct word in English is Plantarflexion.  

Moreover, you have not addressed any of my methodological concerns. It needs to be explained as to why one construct was chosen over the other even if this was just retrospective (which of course introduces a whole different can of worms to the paper and greatly reduces its utility). Again, was a power analysis done apriori or not? Even if retrospective this is important to do. How do you know if you were just simply underpowered to detect a difference?

I do think the discussion is better. You actually focus on the topic of the paper instead of non relevant factors. However, it still requires significant editing.

I have read carefully through the manuscript again, and there are still simply too many English language edits for me to list them all here, this is not my job as a reviewer. This manuscript still requires extensive edits before I will be able to accept it.

Author Response

Thank you for your time in responding to my comments, I think the manuscript has been much improved. However, I still have a number of concerns with this manuscript. Some of my comments you have not addressed at all, for example, in the conclusion you state “internal bracing only showed superiority to knot-tying suture bridge anchoring at the early postoperative stage” and you do not mention that this is only with regards to VAS pain score. I really think that is a result that needs to be greatly qualified, and I am concerned that you did not address this point when I had brought it up on my previous review.

►Response: The authors are grateful to your very detailed instructions and insightful views. The comments really help us a lot in improving our manuscript to reach a possible level for publication. We do learn a lot about scientific writing and study design in the review and revision. We have corrected the last sentence of the conclusion as “However, internal bracing only had less pain compared to knot-tying suture bridge anchoring at the early postoperative stage.”

Furthermore, when you attempted to address my concern about planta flexion, you completely missed my point, the correct word in English is Plantarflexion.

►Response: Thanks for your kind reminder. We have corrected this error.

Moreover, you have not addressed any of my methodological concerns. It needs to be explained as to why one construct was chosen over the other even if this was just retrospective (which of course introduces a whole different can of worms to the paper and greatly reduces its utility). Again, was a power analysis done apriori or not? Even if retrospective this is important to do. How do you know if you were just simply underpowered to detect a difference?

►Response: Actually, we did not choose the surgical methods for these patients. The knotless internal brace technique was only introduced at our hospital in 2017. Before that time, all patients with insertional Achilles tendinopathy were treated with the knot-tying suture bridge technique. After 2017, most cases were treated with the knotless internal brace technique.

        The present sample size is the maximum number that we can achieve. It could be the case that we failed to detect a difference due to an underpowered sample size. This paper may provide data for a future meta-analysis that can make a solid and sound conclusion based upon a large sample by aggregating many studies like this one.

I do think the discussion is better. You actually focus on the topic of the paper instead of non relevant factors. However, it still requires significant editing.

I have read carefully through the manuscript again, and there are still simply too many English language edits for me to list them all here, this is not my job as a reviewer. This manuscript still requires extensive edits before I will be able to accept it.

►Response: Thanks for the suggestion. We have carefully examined the whole manuscript and thoroughly checked the grammar and spelling.